Potential factors promoting the natural regeneration of Larix principis-rupprechtii in North China

Zhao Weiwen
Sun Yanjun syj_yx@126.com
Gao Yufeng
Institute of Soil and Water Conservation Science, Shanxi Agricultural University , Taiyuan , China
Wehenkel Christian
Electronic publication date: 2023 Aug 8
Publication date: 2023
Volume: 11
Electronic Location ID: e15809
Received 2023 Feb 17; Accepted 2023 Jul 7
Copyright: ©2023 Zhao et al.
Copyright year: 2023
Copyright holder: Zhao et al.
License: This is an open access article distributed under the terms of the Creative Commons Attribution License, which permits unrestricted use, distribution, reproduction and adaptation in any medium and for any purpose provided that it is properly attributed. For attribution, the original author(s), title, publication source (PeerJ) and either DOI or URL of the article must be cited.
License URL: https://creativecommons.org/licenses/by/4.0/

Keywords: Natural regeneration, Environmental factors, Plantations, Management implication, Litter layers, Guandi Mountain

Funding: Scientific Research Project of Shanxi Province doctoral work award fund SXBYKY2022146 This work was supported by the Project supported by the Scientific Research Project of Shanxi Province doctoral work award fund (No. SXBYKY2022146). The funders had no role in study design, data collection and analysis, decision to publish, or preparation of the manuscript.

==============================
Natural regeneration plays an important role in species diversity and evolution. Exploring the causes of variation in regeneration dynamics can provide key insights into the factors affecting regeneration. However, the relationship between the regeneration of Larix principis-rupprechtii and environmental factors in North China has remained unexplored. In this study, 14 plots were established based on the three extents of regenerated plant numbers in Shanxi Province. Redundancy analysis determined that environmental factors (topography, stand structure, soil property, and litter) affected natural regeneration. Structural equation modeling identified the most important direct and indirect factors that affected L. principis-rupprechtii natural regeneration. Litter thickness, canopy density, and adult tree diameter at breast height were positively correlated with natural regeneration. Aspect and total nitrogen volume were negatively associated with natural regeneration. Additionally, there was no significant correlation between natural regeneration and other environmental factors (altitude, slope, adult tree height, stand density, soil water content, SOC, total P, available N, available P, or soil enzyme). Further artificial intervention measures should be considered to promote plantation regeneration. These findings provide an effective basis for future forest restorations and sustainable management.

Introduction

Natural regeneration is crucial for the next generation of canopy trees in forest ecosystems (De-Lombaerde et al., 2019). Tree regeneration is typically achieved with close-to-nature management (Fuentes-Montemayor et al., 2021). The regeneration of overstorey trees allows the forest ecological system to retain a higher biomass diversity and ecological quality (Puhlick, Laughlin & Moore, 2012). In most areas, natural regeneration has dominated and replaced artificial regeneration (Puettmann et al., 2015; Ammer et al., 2018). Compared with artificial regeneration, natural regeneration is a more cost-effective and time-saving renewal of a stand using its seeds. It can achieve higher seedling density by making full use of the soil-plant composite system and adapting to complex habitats (Srinivasan, Bhatia & Shenoy, 2015; Kolo, Ankerst & Knoke, 2017).

Promoting the natural regeneration of Larix principis-rupprechtii, one of the main tree species of typical natural secondary forests in the mountains of North China, has been a challenging task. Many researchers have found that L. principis-rupprechtii endures difficult periods when shifting from seeds to seedlings (Hernandez-Barrios, Anten & Martinez-Ramos, 2015; Gao, Li & Zhao, 2020). The priority of these studies was to identify the factors controlling the natural regeneration of trees. The seedling establishment of L. principis-rupprechtii in natural regeneration is more easily affected by abiotic and biotic factors, such as moisture, elevation, soil properties, and temperature (Liu et al., 2012; Terwei et al., 2013). Therefore, a better understanding of the basic characteristics of tree regeneration is a prerequisite for effective management.

Previous studies found that regeneration significantly affects stand density, canopy density, and herb, shrub, and litter layers (Royo & Carson, 2008; Maciel-Nájera et al., 2020; Nakhoul et al., 2020). Stand density and canopy density can influence herb and shrub cover. As herbs and shrubs grow, the production of litter increases, affecting solar radiation. For instance, low understory light levels limit seedling survival, while light availability exerts pressure on shade-tolerant trees (Sangsupan et al., 2021). Litter impacts the composition of species communities by changing the microclimate needed for seeds to germinate (Hu et al., 2016). In turn, litter decomposition affects nutrient recycling. In addition, topography affects runoff, soil drainage, and soil property variation, thus adjusting regenerated conditions (Wang et al., 2001). Strong ties exist between topographic position, soil nutrients, and plantation regeneration (Dessalegn et al., 2014). Previous studies used traditional multivariate analysis to explore the relationships between topography, soil, shrubs, and herbs in order to determine the main determinants of regenerated seedlings (Wang et al., 2016). Although some studies discussed several factors affecting regeneration, few have comprehensively assessed the relevance of potential factors affecting L. principis-rupprechtii regeneration (O’Brien et al., 2007; Caquet et al., 2010; Zhang et al., 2011; Ibáñez et al., 2015).

This study focused on all potential environmental factors influencing the regeneration of L. principis-rupprechtii selected from a few areas in North China. A region of L. principis-rupprechtii natural regeneration in Shanxi Province was selected in order to identify its control factors. Three plantations with different levels of regeneration density were selected for examination: (1) regeneration density ≤ 3,000 tree/ha; (2) 3,000 <regeneration density ≤ 5,000 tree/ha; and (3) regeneration density >5,000 tree/ha. This study was designed to: (i) assess a variety of potential factors affecting L. principis-rupprechtii regeneration, (ii) identify the main factors affecting natural regeneration, and (iii) propose some management measures to promote a level of L. principis-rupprechtii natural regeneration. The results are expected to provide some ideal guidance for managing L. principis-rupprechtii natural regeneration and achieving sustainable development of forest regeneration.

Material & methods

Study site

This study was conducted in the Chailugou region of Guandi Mountain, Jiaocheng County, Shanxi Province (111°28′–111°33′E, 37° 48′–37°51′N) (Fig. 1) (Yang et al., 2017). The annual average temperature is 4.3 °C, with minimum and maximum temperatures of −11.9 °C and 30.7 °C, respectively. It has a temperate continental monsoon climate. The annual average precipitation and evaporation are 822 mm and 1,268 mm, respectively, and the elevation is 1,500–2,831 m (Zhao et al., 2021). The main soil types are mountain cinnamon soil, cinnamon soil, and brown soil (Yang et al., 2017).

Figure 1 The study site location in Jiaocheng County (A). (B) Sampling plots (1–14) in Guandi Mountain, gray boxes indicate the 14 sampling areas.

The dominant plant species are forest trees (L. principis-rupprechtii, Pinus tabuliformis, Picea wilsonii, Picea meyeri, Picea asperata, and Populus simonii), shrubs (Acer tataricum subsp. ginnala, Rosa multiflora, Spiraea salicifolia, Clematis florida, and Berberis amurensis), and grasses (Fragaria orientalis, Geranium wilfordii, Chquuuujrysanthemum chanetii, Lathyrus humilis, Rubia cordifolia, Bupleurum smithii, and Thalictrum aquilegiifolium).

Experimental design

From July to August 2022, an area of 1 km2 in the artificial forest (no thinning or management in 30 years) on Guandi Mountain was investigated to determine L. principis-rupprechtii natural regeneration. The 14 sampling plots (20 × 20 m) were randomly established, and each plot contained regenerated plants and adult trees. The trees within the studied area were surveyed to record height, diameter at breast height (DBH, measurement of the diameter of the tree 1.30 m above the ground), canopy cover, species, and numbers. All trees were classified into three types in each plot (regenerated plants of seedlings: height ≤ 1 m; saplings: height >1 m, DBH ≤ five cm; adult trees: DBH >5 cm), and also according to regeneration degree (high: >5,000 tree/ha; medium: 3,000–5,000 tree/ha; low: <3,000 tree/ha). Each plot was set with a hand-held global position system (GPS) to measure longitude, latitude, and elevation. Growth cores were used to determine the age of adult trees, while a compass was used to record aspect (direction of projection of slope normal on horizontal plane) and slope (degrees) (Table 1).

Data collection

The research site was located in the rocky mountain area, which was near the bedrock, 60 m deep, and very dry. Five soil sampling plots were collected with an X-shaped collection scheme at soil depths of 0–20 cm, 20–40 cm, and 40–60 cm using a soil auger after three days without a rainfall event. This was replicated three times. These three soil layers were mixed and placed in an aluminum box as composite soil samples and brought back to the laboratory for drying and weighing (105 °C, 24 h) to calculate soil water content (SWC). Meanwhile, the dried soil samples were crushed and sieved with a 1.5 mm screen to remove rocks and roots. Soil organic content (SOC) was measured using H2SO4-K2CrO7, total nitrogen (TN) was determined using the Kjeldahl method, total phosphorus (TP) was determined using the colorimetric method after digestion with HClO4-H2SO4, and available nitrogen (AN) was measured using the method shown in Li et al. (2018). Furthermore, soil enzymes (sucrase, phosphatase, and urease) were measured following Ping et al. (2021).

Table 1 Summary of the stand characteristics in sample plots.

Plot	Basal area (m2 ha −1)	Altitude (m)	Longitude	Latitude	Stand density (Tree ha −1)	Aspect	Slope (°)	DBH (cm)	Height (m)	Age (a)	
1	3.89	2,063	11132 ′32″	3751 ′21″	175	N-W	19.4	44.5	21.13	56	
2	2.50	2,066	11132 ′33″	3751 ′21″	125	N-W	13	35.7	23.9	59	
3	1.95	2,060	11132 ′28″	3751 ′19″	250	N-W	23	31.49	17.81	50	
4	2.39	2,070	11132 ′31″	3751 ′20″	225	N-W	15.7	34.87	20.87	64	
5	1.94	2,078	11132 ′26″	3751 ′17″	75	N-W	17	31.47	21.73	64	
6	1.14	2,029	11132 ′25″	3751 ′17″	175	N-W	25	24.14	19.84	62	
7	2.07	2,018	11132 ′24″	3751 ′15″	125	N-W	22	32.48	16.54	64	
8	1.75	2,002	11132 ′19″	3751 ′14″	75	N-W	24	29.87	21.8	55	
9	3.51	2,007	11132 ′18″	3751 ′15″	100	W	22	42.3	21.1	82	
10	1.64	1,990	11132 ′19″	3751 ′15″	100	W	32	28.88	21	70	
11	0.75	2,022	11132 ′25″	3751 ′20″	50	W	15	19.6	12.3	19	
12	2.36	2,025	11132 ′26″	3751 ′20″	50	W	18	34.7	14.5	65	
13	2.83	2,023	11132 ′24″	3751 ′20″	50	W	21	37.95	24	19	
14	4.67	2,000	11132 ′23″	3751 ′21″	75	W	8	48.77	22.7	59	
Notes.

Note: N-W is the northwest aspect of each plot, and W is the west aspect of each plot.

Sucrase: 5 g of air-dried soil was mixed with 15 mL of 8% concentration sucrose solution. Then, 5 mL of phosphate buffer solution (pH 5.5) and five drops of phenol were added. After culturing at 37 °C for 24 h, we took 1 mL of supernatant and added 3 mL of a 3, 5-dinitrosalicylic acid solution. The released sugar was measured at 508 nm in the spectrophotometer.

Phosphatase: 5 g of air-dried soil and five drops of toluene were mixed. We added 20 mL of 0.5% sodium diphenyl phosphate and shook well. Then, the mixture was incubated at 37 °C for 2 h. The phenol released was measured at 510 nm in the spectrophotometer.

Urease: 5 g of air-dried soil and one mL of toluene were mixed. After standing for 15 min, 10 mL of 10% concentration urea solution and 20 mL of citric acid buffer solution (pH 6.7) were added. Then, the mixture was incubated at 37 °C for 24 h. After filtration, 1 ml supernatant was transferred into a volumetric flask. Then we mixed it with 4 mL of sodium phenolate and 3 mL of sodium hypochlorite. After standing for 1 h, the ammonia released was measured at 578 nm in the spectrophotometer.

After sampling the soil, we established five herb quadrats (1 × 1 m) and five shrub quadrats (5 × 5 m) and used an X shape sampling method in these plots. At each quadrat, the species, numbers, height, and coverage were determined. The five litter samples were also collected near herb and shrub sample plots with an X-shape pattern to record their thickness, and then they were bagged and weighed. The collected litter was taken to the laboratory for drying and weighing (85 °C, 24 h) and the accumulation of litter was calculated (Dong et al., 2021).

Data analysis

Natural regeneration data (regenerated numbers, tree DBH, and tree height per plot) could not be standardized using various methods. Therefore, the effects of treatments via nonparametric tests (Kruskal-Wallis ANOVA) were analyzed. The raw data were ranked to test the interaction between multiple factors. In this study, the number, DBH, and height of regenerated plants were used to represent the response variable. Environmental factors were divided into four categories to represent the explanatory variable: topographic factors (altitude, slope, and aspect), stand structure (stand density, adult tree average DBH, adult tree height, and canopy density), soil properties (soil water content, SOC, TN, TP, available phosphorus, available potassium, ammonia nitrogen, urease, sucrase, and phosphatase), and litter (thickness and accumulation).

To measure the factors’ contribution rates, the environmental gradients were identified using CANOCO (version 5.0). Preliminary detrended correspondence analysis of the regenerated sampling plots demonstrated that the gradient lengths were < 3.0, indicating that the regeneration plots showed linear responses to latent environmental variation. This was evidence of the reasonability of the linear multivariate methods (Li et al., 2018). Therefore, redundancy analysis (RDA) was used to determine the community’s predominant environmental factors. The scale of the ordination centered on correlations among the species, and Monte Carlo simulations with a significance level of 99% were used to test the statistical significance based on random permutations (Chen & Cao, 2014). The regeneration data were transformed through logarithmic transformation to reduce the influence of extreme values in the RDA (Gazer, 2011). In order to test the relationships between environmental factors and regeneration, all values for environmental factors and regeneration were square-root transformed to ensure the uniformity of variance before statistical analysis. The transformed values were analyzed using the ANOVA method, and simultaneous Pearson correlation coefficients were determined to test the degree of relationships between environmental factors and regeneration. Statistical parameters and tests were conducted using SPSS 22.0 (Chicago, IL, USA) and graphics were drafted in Origin 2023 (Northampton, MA, USA).

To test all potential effects of environmental factors on regeneration, structural equation modeling (SEM) was established by transforming the data sets into a path relation network (Malik et al., 2018). A confirmatory approach was used to measure the maximum likelihood of data fitting the hypothesized path model and deduce the environmental factors influencing natural regeneration. The fitting of the path model and the relationship between regeneration and environmental factors were verified using the Lavan R (version 4.2.1) package (Rosseel, 2012; Team CR, 2015). In the baseline comparisons, the fittest model was identified by a high comparative fit index(1 >CFI >0.9), high goodness of fit index(1>GFI >0.9), low root mean square error of approximation index (RMSER ≤ 0.08), and low standardized root mean square residual index (SRMR <0.08).

Results

Characteristics of natural regeneration

Natural regeneration was abundant in the study area after our investigation. A great number of regenerated saplings (Nsa) were recorded (Table 2), particularly in maximum plot 4 (552 saplings) and minimum plot 14 (Nsa = 26). However, the number of regenerated seedlings (Nse) was low except in plot 1 (Nse = 62), plot 4 (Nse = 68), and plot 10 (Nse = 19). Additionally, there was a low number of adult trees (between 1 and 10). The population of regenerated saplings made up more than 80% except in plot 1 (59%), while regenerated seedlings and adult trees were less occupied in each sample plot (Table 2). However, there were no regenerated seedlings recorded in plots 8 and 11. Sampling plots 9 (density = 1,675 tree/ha), 11 (density = 1,525 tree/ha), and 14 (density = 700 tree/ha) were recorded with low regeneration degrees. Other sample plots with a high regeneration degree of L. principis-rupprechtii contained more than 3,000 tree/ha.

Table 2 The Larix principis-rupprechtii density of recruited trees, saplings, and seedlings in study plots.

Density (Tree ha −1)	Study site	
	1	2	3	4	5	6	7	8	9	10	11	12	13	14	
Seedling	1,550	175	425	1,700	125	50	75	0	50	475	0	100	150	50	
Sapling	2,450	5,775	7,625	13,800	5,800	9,825	11,150	2,625	1,625	2,925	1,525	7,850	4,200	650	
Adult tree	175	125	250	225	75	125	125	75	100	100	25	50	50	75	

Environmental factors for RDA analysis

RDA determined the relative contributions of topographic factors, stand structure, soil properties, and litter variables on L. principis-rupprechtii regeneration. The correlation between multivariate variables was evaluated by determining the best predictive factors for vegetation regeneration based on statistical theory. Figure 2 shows the RDA ordination diagram. The correlation between the corresponding variables is represented by the cosine values of the environment variables in the graph, the sine cosine value represents the positive correlation between the variable, and the negative cosine value represents the negative correlation.

Figure 2 Ordination diagram of the redundancy analysis (RDA) results of topographic factors, stand structure, soil properties, and litter in sample plots.

(A) The relationship between the number (Re-Number), DBH (Re-DBH), height (Re-Height), and impact factors of regenerated trees; (B) the relationship between sample plot distribution and impact factors. The direction of the arrows corresponds to the correlation (positive or negative) among the environmental factors with the axis. The lengths of the arrows indicate the extent of correlation which a factor impacts regeneration and longer lines indicate further correlations. Abbreviations of stand structure, soil properties, and litter variables are as follow: StaD, stand density; Adt-DBH, adult tree average diameter at breast height; Adt-Height, adult tree height; CanD, canopy density; SWC, soil water content; TP, total phosphorus; TN, total nitrogen; AN, ammonia nitrogen; AP, available phosphorus; LitT, litter thickness, and LitA, litter accumulation.

The results show a strong correlation between regeneration and environmental factors as a 94.2% variation in regeneration is shown on axis 1 (Fig. 2). Figure 2B shows all biplots of the RDA analysis. Litter thickness, litter accumulation, stand density, and canopy density positively correlated with the regenerated number, while altitude, soil water content, and total P had a weak correlation with the regenerated number (Fig. 2A). Litter thickness and altitude were the strongest and weakest effects, respectively. The variables of TN, available P, aspect, adult tree height, and adult tree DBH negatively correlated with the regenerated number, and aspect had the largest negative effect. Regenerated height strongly positively correlated with aspect and negatively correlated with litter accumulation, litter thickness, stand density, canopy density, total P, soil water content, and altitude (strong negative correlation), while adult height, adult tree DBH, TN, available P, and available N had a weak effect. Regenerated tree DBH positively correlated with aspect and available N, the latter being the strongest. Additionally, it negatively correlated with other environmental factors, with altitude having the strongest negative effect. Sample plots 2, 3, 4, 6, 7, and 12 showed the same patterns except for plot 5, while plots 1, 8, 9, 10, 11, 13, and 14 showed the same patterns and represented the largest degrees of natural regeneration, with sample plot 4 showing the highest (Fig. 2B).

In the RDA analysis, litter thickness and canopy density contributed more than 10% (Table 3). The RDA1 axis positively correlated with total P, soil water content, altitude, stand density, canopy density, litter thickness, and litter accumulation, while available N, available P, TN, aspect, average adult tree DBH, and adult tree height showed a negative correlation (Table 4). Litter thickness had a greater positive correlation coefficient with RDA1 (0.696), while available N had the lowest negative correlation coefficient with RDA1 (−0.134). Aspect was most highly correlated with RDA2 (0.522), followed by available N (0.055). Other variables showed negative correlations.

Table 3 Percent variance of environmental factors affecting regeneration.

Factor	Explains (%)	Pseudo-F	P	
LitT	60.4	18.3	0.004	
CanD	20.5	11.9	0.004	
Altitude	3.7	2.4	0.12	
AP	2.5	1.7	0.182	
Aspect	1.3	0.9	0.402	
SWC	2.8	2.3	0.14	
TN	2.7	2.7	0.126	
Adt-DBH	1.2	1.2	0.324	
Adt-Height	2.1	2.1	0.138	
StaD	1.2	2.1	0.164	
TP	0.9	2.8	0.17	
LitA	0.6	5	0.122	
AN	0.1	<0.1	1	
Notes.

LitT litter thickness

CanD canopy density

AP available phosphorus

SWC soil water content

TN total nitrogen

Adt-DBH adult tree average diameter at breast height

Adt-Height adult tree height

StaD stand density

TP total phosphorus

LitA litter accumulation

AN ammonia nitrogen

Table 4 Coefficients for environmental factors for RDA1 and RDA2.

Environmental factor	RDA1	RDA2	
Total P	0.120	−0.278	
Available N	−0.134	0.055	
Available P	−0.382	−0.292	
Total N	−0.394	−0.226	
Soil water content	0.401	−0.592	
Altitude	0.496	−0.692	
Aspect	−0.561	0.522	
Stand density	0.616	−0.114	
Canopy density	0.547	−0.108	
Litter thickness	0.696	−0.203	
Litter accumulation	0.416	−0.008	
Average adult tree diameter at breast height	−0.215	−0.144	
Average adult tree height	−0.150	−0.096	

Regeneration correlation with environmental factors

According to RDA analysis, 13 environmental factors were selected. Figure 3 shows the correlation between stand structure, soil properties, litter variables, and topographic factors. The number of regenerated trees were positively correlated with stand density (0.62) and canopy density (0.55), and significantly positively correlated with litter thickness (0.70). The extent of regeneration (including number, DBH, and height) was positively correlated with litter thickness (0.70, 0.69, and 0.55, respectively). Soil water content (−0.72) and altitude (−0.85) were significantly negatively correlated with tree regeneration height. Soil water content was notably negatively correlated with aspect (−0.81) but significantly positively correlated with stand density (0.68) and positively correlated with altitude (0.66). TN was greatly positively correlated with available N (0.67). Litter thickness was negatively correlated with aspect (−0.64) but positively correlated with stand density (0.57) and litter accumulation (0.61). Adult tree DBH was negatively correlated with litter thickness (−0.60).

Figure 3 Correlation analysis of environmental factors and regeneration.

Re-number, regenerated tree number; Re-DBH, regenerated tree diameter at breast height; Re-Height, regenerated height; TP, total P; AN, available N; TN, total N; SWC, soil water content; StaD, stand density; CanD, canopy density; LitT, litter thickness; LitA, litter accumulation; Adt-DBH, adult tree diameter at breast height; Adt-Height, adult tree height. Note: The numbers above diagonals indicate Pearson correlation values “r”.

Structural equation model (SEM)

Based on maximum likelihood estimation, eight environmental factors were selected to form the SEM. All indexes were high in this model, which had a great degree of fit (χ2/ df = 0.446, P = 0.874). The comparative fit index (CFI) and goodness of fit index (GFI) were 0.999 and 0.991, respectively. The parsimony goodness of fit index (PGFI) and parsimony normed fit index (PNFI) were 0.505 and 0.552, respectively. The standardized root means square residual (SRMR) and root mean square error of approximation (RMSEA) were 0.027 and 0.002, respectively.

According to Fig. 4, litter thickness and regeneration were positively associated, with a path coefficient of 0.87. Regeneration was positively correlated with canopy density, with a path coefficient of 0.63. Altitude was positively correlated with litter thickness, canopy density, and soil water content, with path coefficients of 0.25, 0.71, and 0.31, respectively. Aspect was positively correlated with litter accumulation, with a path coefficient of 0.31. TN was positively correlated with litter thickness, with a path coefficient of 0.23. Aspect and canopy density were positively correlated, with a path coefficient of 0.30. Meanwhile, litter accumulation was positively associated with litter thickness, with a path coefficient of 0.81. Regeneration was negatively associated with litter accumulation, TN, and aspect, and the path coefficients were −0.08, −0.17, and −0.18, respectively. TN and litter accumulation were negatively correlated, with a path coefficient of −0.47. Aspect was negatively associated with litter thickness and soil water content, and the path coefficients were −0.40 and −0.53, respectively.

Figure 4 Corrected structural equation model with standardized path coefficients between influencing factors and regeneration.

Figures on the arrows indicate standardized path coefficients. Red arrows indicate negative effects, while blue arrows indicate positive association.

Discussion

According to SEM analysis, litter thickness, altitude, aspect, canopy density, adult tree DBH, and TN were selected as factors that affected L. principis-rupprechtii natural regeneration.

Effect of topography on regeneration

Soil is essential to the growth and distribution of vegetation. Soil and vegetation are affected by topography (Liu et al., 2012; Parker, 2013). Topographic factors can affect seeds, water, and nutrient redistribution. For instance, aspect has a significant effect on plant community structure and the extent of natural regeneration (Fu et al., 2004; Toure, Ge & Zhou, 2015; Wang et al., 2016; Maciel-Nájera et al., 2020). Previous research results indicated that aspect and altitude were the major topographic factors affecting regeneration. However, SEM analysis indicated that altitude did not influence regeneration. In this study, the difference between high and low elevations in the sampling plots was 88 m. Many studies found little effect on the development of vegetation and regeneration when the elevation change was <300 m (Wang et al., 2006; Liu et al., 2012), which may not be enough to cause variations in water conditions. Therefore, elevation was not chosen as a variable affecting regeneration. Aspect was positively related to the activity of aboveground organisms and distribution of herbs, and was negatively correlated with regeneration (Vitousek et al., 1994; Scowcroft, Turner & Vitousek, 2010). There were two kinds of aspects (northwest and west) recorded in our study site, which determined the amount of accepted solar radiation (Sariyildiz, Anderson & Kucuk, 2005). L. principis-rupprechtii seedlings need enough light to grow early-stage from seeds to seedlings then saplings. The northwest aspect received more solar radiation than the west aspect. A great regeneration density was recorded in the northwest aspect in this study. Soil temperature and soil water availability are controlled by aspect, which in turn affect seedling establishment and growth (McNab, 1992; Zhang et al., 2012). Topography can affect soil depth, profile development, and the accumulation of soil nutrients, which indirectly influence the distribution of plants and species composition (Sariyildiz, Anderson & Kucuk, 2005; Dessalegn et al., 2014; Liang & Wei, 2020). In this study, regeneration was negatively correlated with aspect (−0.18), suggesting that it was easier for seeds to accumulate and grow in the northwest aspect with more solar radiation.

Effect of stand structure on regeneration

Stand structure greatly influences the level of regenerated seedling and plant restoration, which can affect seedling richness and density (Liu et al., 2020). This study investigated the effect of stand structure on natural regeneration. The effect of different factors depended on the life stages of the regenerated seedlings. One of the structure factors, canopy density, affects stand structural composition, litter cover, species richness, and composition (Takahashi & Mikami, 2008; Chen & Cao, 2014). Regenerated seedlings need light to grow, so it is important to increase light capture during regeneration. Seedlings need to reduce self-shading to accept more efficient light in the forest understory (Takahashi & Mikami, 2008). The SEM analysis suggested that adult tree DBH was positively correlated with natural regeneration. Mother trees with greater DBH typically obtained more soil water, light, and other resources to support seedling growth, and occupied a dominant position in the forest (Li et al., 2005). Many seeds produced by robust mother trees were higher in quantity and quality, which might explain the high regeneration density in the study area. There was a positive correlation between canopy density and regeneration, which was consistent with the results of Ali et al. (2019) and Li et al. (2012). Meanwhile, the strength of this correlation was higher than that of adult trees’ DBH. The regenerated period of light requirements for growth increased with the increasing canopy density. Shortage of light supplements could cause poor growth and even death (Gaudio et al., 2011; Wang et al., 2017). Some studies have also indicated that light is one of the most important environmental factors influencing the development of natural regeneration. Direct light transmission is more conducive to seed germination and seedling growth during natural regeneration (Canham et al., 1990; Kneeshaw, Kobe & Messier, 2006). In general, the relationship between canopy density and adult tree DBH on L. principis-rupprechtii regeneration in this study was highly consistent with the findings of other studies (Ares, Neill & Puettmann, 2010; Koorem & Moora, 2010; Ali et al., 2019; Liang & Wei, 2020).

Effect of litter on regeneration

Understory shrubs, herbs, and litter play an important role in maintaining the biodiversity of local forest ecosystems and affecting natural forest regeneration (Li et al., 2018). As shrub height and herb coverage increase, more light is intercepted, and more litter is accumulated. In turn, the availability of space for growth decreases (Facelli & Pickett, 1991; Caccia & Ballaré, 1998; O’Brien et al., 2007). In this study, shrubs played a negligible role, were less distributed in sampling plots, and had no effect on regeneration. In addition, L. principis-rupprechtii is a shade-tolerant species, so there was no clear effect on regenerated seedlings (Vayreda et al., 2013; De-Lombaerde et al., 2019). However, some studies have suggested that limitations on understory vegetation can promote natural regeneration since the herb can provide a shady environment to support the growth of seedlings. In turn, seedlings restrain the growth of herbs, which explains the existence of vegetation in a few understories (Bose et al., 2012; Pamerleau-Couture et al., 2015; Boivin-Dompierre, Achim & Pothier, 2017; Skay et al., 2021).

RDA analysis suggested that litter thickness could account for 60.4% for regeneration. SEM analysis showed that it had a path coefficient of 0.87 for regeneration. Therefore, litter thickness was closely correlated with regeneration. Generally speaking, the seeds of L. principis-rupprechtii were primarily scattered in soil layers and they measured greater than 2 mm in length (Liang & Wei, 2020). As the thickness of litter layers increased, the water holding capacity increased. Stable temperatures and better water holding capacity provide suitable growth conditions for germination and limit soil water evaporation (Spanos, Daskalakou & Thanos, 2000; Boydak, 2004; Petrou & Milios, 2020). Some studies have indicated that seeds do not grow and even die in litter layers since the radicles fail to reach the soil layers to attain adequate nutrients. Additionally, the litter has an auto-toxic effect on seed germination and seedling growth (Pardos et al., 2007; Willis et al., 2021). Nevertheless, in this study, litter thickness had positive correlations with regeneration (Fig. 4) since the adequate litter thickness and stable decomposition rate of litter could reduce mechanical and physiological barriers to promote natural regeneration and help seeds find favorable conditions to germinate (Eckstein & Donath, 2005; Baker & Murray, 2010). Therefore, the litter thickness was in an ideal state in this area and can be adjusted in future management to achieve the optimal regeneration density.

Effect of soil properties on regeneration

Soil nutrient circulation is an important factor affecting natural forest regeneration (Chen & Cao, 2014). Seed germination and seedling growth require soil resources, such as SOC, soil water content, TN, and total phosphatase (Will et al., 2005). SEM analysis showed that TN greatly contributed to natural regeneration compared with other soil properties. Numerous litters were accumulated and decomposed in the surface soil layer in extensively managed forests, increasing TN content. TN was positively correlated with canopy density and negatively correlated with regeneration, while litter thickness was positively correlated with TN. Hence, species were deduced to increase TN, which was in agreement with the results of Baker & Murray (2010) and Li et al. (2015). In addition, TN is one of the most important variables that impacts species richness and plant distribution (Zuo et al., 2012). Meanwhile, more nutrients were released from inputted litter with forest regeneration (Deng et al., 2013). Nevertheless, as the TN content decreased, the number of species increased (Rhoades et al., 2009; Liu et al., 2011). Through observation, the richness (number) and quantity (height and DBH) of species varied in the RDA, indicating insufficient regeneration quality in the study site. Though high TN volume can restrict natural regeneration, a great proportion of TN consumption is needed for regenerated seedlings (Xu et al., 2018). Therefore, artificial restoration is necessary for the later stage.

Management implications

Environmental factors (aspect, canopy density, adult tree DBH, litter thickness, and TN) and the regeneration of L. principis-rupprechtii were mutually related and restricted. Improving soil quality to promote regeneration should be considered. Organic soil formation is a process that is accomplished in parallel and in synergy with successful L. principis-rupprechtii regeneration. Effective implications should be considered, such as the appropriate removal of litter to decrease thickness in order to promote seed germination. Scattered seedlings and dead trees should be removed to increase the distance between seedlings and adult trees to allow more available light to reach L. principis-rupprechtii.

Conclusion

A better understanding of the environmental factors affecting seedling growth in order to achieve L. principis-rupprechtii regeneration was needed. Based on RDA and SEM analysis, litter thickness was the most important variable affecting natural regeneration, followed by canopy density, adult tree DBH, aspect, and TN. Greater canopy density, adult tree DBH, low TN and distribution of the northern-west aspect were beneficial for regeneration. These proposed intervention measures should be adopted in forest management, including reducing thickness in thicker areas and removing dead trees to increase the forest gap. We suggested that some aggregate regenerated seedlings near the adult trees were properly removed to the northwest aspect. Our results may contribute to better management of regeneration and sustainability for forest plantations in the North China mountain region.

Supplemental Information

Data S1 The basic information for each sampling plots, analysis for RDA and for SEM

Click here for additional data file.

We thank the anonymous reviewers for their useful comments.

Additional Information and Declarations

Competing Interests

Author Contributions

Data Availability

The authors declare there are no competing interests.

Weiwen Zhao conceived and designed the experiments, performed the experiments, analyzed the data, prepared figures and/or tables, authored or reviewed drafts of the article, and approved the final draft.

Yanjun Sun conceived and designed the experiments, performed the experiments, authored or reviewed drafts of the article, and approved the final draft.

Yufeng Gao conceived and designed the experiments, performed the experiments, prepared figures and/or tables, and approved the final draft.

The following information was supplied regarding data availability:

The raw data is available in the Supplemental Files.

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
