# Peer review of "Potential factors promoting the natural regeneration of Larix principis-rupprechtii in North China"

_PeerJ, doi:10.7717/peerj.15809_

## Round 0.1 · original submission · Major Revisions

The manuscript is confusing because the English is not clear. The authors exaggerate a bit with statistical analysis, but at the same time, they do not present clearly their statistical support and results. Somehow you present supported results but the manuscript is hard to follow.

Reviewer 1 ·

Basic reporting

1) Literature references, sufficient field background/context provided: Regular
2) Professional article structure, figures, tables. Raw data shared. Yes
3) Self-contained with relevant results to hypotheses: Unclear
In general, the Introduction could be improved by citing more works related to natural regeneration. The overall structure of the article is good. The figures and tables are sufficient.

Experimental design

1) Original primary research within Aims and Scope of the journal. Yes
2) Research question well defined, relevant & meaningful. It is stated how research fills an identified knowledge gap. Regular

The knowledge gap being investigated is not clearly identified and no statements are made about how the study contributes to filling that gap. On the other hand, there is no defined research question, although the manuscript has three purposes.
Regarding the third point, which states: “propose some management implications to promote the level of L. principis-rupprechtii natural regeneration", there is no description of how this point is covered in Methods.

3) Rigorous investigation performed to a high technical & ethical standard. Ethical yes. But technical, no.
Methods described with sufficient detail & information to replicate. No

There are two serious aspects that the authors should address:
1) The sampling design should be described and justified.
I did not find a description of the sampling method used to establish the 14 plots (20 x 20 m). Similarly, it is not clear how the five soil sampling plots were established; were these five plots selected from within the 14 plots or are they independent plots?
In sum, how do the authors ensure that there was representativeness?
2) Clarify the analyses chosen. The authors should clearly state (a) what was done, (b) how a particular analysis was done (steps followed), (c) why it was done. Linking with the objectives of their work; and, (d) how they evaluated the models performed.

Validity of the findings

1) All underlying data have been provided; they are robust, statistically sound, & controlled: Moderately
2) Conclusions are well stated, linked to original research question & limited to supporting results.
Should be improved

Additional comments

SPECIFIC ASPECTS

L.95-96. How many are "last years?" 10 years, 50 years, two years? Please specify.
L. What sampling method did you use to establish the 14 plots with 20 x 20 m area. This is the most important information in an investigation. When the sampling method is not appropriate, there can be bias in the collection of information and therefore bias in the results.
L.98. Although DBH is a common measurement in forestry, it is necessary to inform other readers who are unaware of this standard measurement. Clarify if it is 1.30 m above ground level?
L. 102. What is GPS?
L. 103 and 104. Mention the error of measuring devices. Accuracy. The authors should clearly describe how the measurements were recorded. The slope and exposure in degrees or percentages?
L. 117 Please describe briefly about the method of method of Xiangping et al. (2021).
L. 124 The description of data analysis has no order. Please mention in order how things were done and for what reason.
L. 125. What do the authors mean by "normalized". Standardized? Or do they mean that the data did not follow a Gaussian distribution?
L. 129. Explain what the "sorting analysis" consists of and for what purpose it was performed.
L.129-133. The paragraph is very confusing. Clarify and make it understandable.
L. 133-135. Check this wording, please : “redundancy analysis (RDA)” was carried out in the CANOCO 5.0 software to determine the community’s predominant environmental factors? The Redundancy analysis is a method of analysis.
L.134-137 Short sentences seem not to be linked completely. The most important for the audience, that is; how it was done and why it was done.
L. 137-138. The authors write: “to reduce the influence of extreme values”. Where, in which method of analysis, were there outliers and outliers? Please mention.
l. 140. About: “to represent the species” To represent where?, there were more species? How do the authors refer to representation? Or "represent" is misused here. Review, please.
L.45-147. The authors write: “All values for these factors were square-root transformed.” They refer to the properties of the soil? Then they say: All values after logarithmic transformation were analyzed using. This is confusing. First they take square root and then loragithm? Or how.
On the one hand, the description is very confusing. On the other hand, it seems that the authors are forcing the data to use certain methods of analysis. That is, instead of looking for the most appropriate analysis method, they manipulate the data in order to use an analysis method that they have already decided to use.

L.147. Provide the significance level used to reject the null hypothesis instead of saying: p < 0.01 or p < 0.05.
L. 148. “Pearson correlation coefficients” indicates the degree of linear correlation between two or more variables.
It gives the impression that the authors are unfamiliar with the methods of analysis and their rationale. They should make clear to the audience, what they did, how they did it, why they did it and where they did it (Software). For example, what is Structural equation, why was it used? What part of the research question does it answer? How was it done step by step?
L. 156 Clearly state what it consists of: “The fitting of the path model and the relationship between structure”.
L. 158. “high goodness of fit index”. What does it consist of?

l.- 164. “Natural regeneration was better in the study area (Fig. 2, Table 2) after field investigation” The authors imply, if the research had not been done, the study area would not be better off?
L. 176. “Based on statistical theory, the best predictors” Statistical theory does not define predictors. Please use the most appropriate expressions. Never. Perhaps indicator, significance index, coefficient, etc-.....
L. 179. The following does not refer to results but to materials and methods: “meanwhile, the correlation between variables and multivariable data was evaluated.”

L. 243: L. 243: Instead of writing "According to a series of analyses"... the authors should clearly state which analysis, according to the coefficient of. The index of importance showed that ..... Then put a table or a figure where the audience can verify their statements.

Finally, much of the discussion is based on the analysis of field conditions. In that sense, the study becomes more relevant. For example, the thickness of the leaf litter, which is based on field observation.

Reviewer 2 ·

Basic reporting

The English writing is confusing in several sentences, preventing to interpret unambiguously what was the intended idea. See my specific comments below

Literature referenced is OK, although it seems that in some references it was confounded the given name with the family name.

Some Figures and Tables are a bit confusing and redundant. See my specific comments below.

It provides relevant results, although clarity need to be improved.

Experimental design

The manuscript falls OK into the scope of the journal.
Research question is defined (what environmental and stand conditions favor the Larix principis-rupprechtii regeneration).
Acceptable methodology although its description is confusing. See my specific comments below.

Validity of the findings

The findings are supported by statistical analysis, although the description of that is confusing on some parts. See my specific comments below.
Conclusions description need to be improved. See my specific comments below.

Additional comments

Specific comments
Abstract
Lines 15-16. It is not clear what you mean with “extents” on “… the three extents of
regenerated L. principis-rupprechtii…” Do you mean regions where it distribute naturally?
Introduction
L. 35. “Natural regeneration is replacing artificial planting in most places in a dominant manner” ??? where is happening that? In China? OK, I do not know; if it is so, say that explicitly. Or are you making a generalization ? Well, I do not see that trend worldwide.
L. 36. What do you mean with “traditional regeneration”? Traditional reforestation? It is confusing because next sentence you tal about “natural regeneration”
L. 67. It is confusing “Three plantations regenerated degree were classified”. Perhaps better: “Three plantations with different levels of regeneration density were identified tobe examined” or something like that.
L. 69. “assess the all-potential factors affecting”. Well, it is hard to know in advance what are ALL the affecting factors. Perhaps to say: “assess a variety of potential factors affecting..”
70-71. “propose some management implications” is unclear. Perhaps “propose some
management actions? Measures?

Methods
81-83. I do not seed the need or utility to provide the atmospheric pressure. I believe that with temperature, precipitation and elevation suffice.
87. “The dominant plant species were forests”. Do you mean “The dominant plant species were forest trees? Because the you talk about shrubs and grasses.
95-96. “Then the trees within the plot wood were seized feet” is confusing. “Plot” or do you mean “studied area”? What is “seized feet”?
137-138. “were used to represent the species”. No clear what you mean with “to represent the species” What species? Larix only? Did you focused only on Larix regeneration? If so say that. To represent the Larix regeneration as response variable or what?
148. What do you mean with “drafted in”? Graphics were done with Origin?

Results
161. “Natural regeneration was better in the study área” It was better than what? Trhen previous measurements? Previous published studies? Or do you mean that in general regeneration was abundant in general?
162-164. It is better y you present those results (here in the text and on Figure 2) as number of recruited trees/saplings/seedlings or the total per Ha, rather than the number of individuals in the plot, because the reader might not have present the size of the plot to valuate if 552 or 26 is too much or too low.
Figure 4. Nowhere in the Figure heading indicate what are the numbers above diagonals (Pearson correlation values “r” ?). Same thing for symbols. What is the difference between a circle and an ellipse or something between a circle and an ellipse?
224-235- All that info is difficult to follow. It might be good to find a different way to present so many values. Perhaps a table where it is sorted from a high to a low value of correlation of the variables, and divide that in sections of significant or no significant or so?
Table 2 is redundant of Figure 2. See also my comment for how to present the results.

Discussion
245 – 247. It sounds contradictory. Altitude affects or not regeneration?
256-257 “The Northwest aspect could receive more solar radiation than the West aspecto” ???? Usually northern aspects are the ones that receive less light! At least on the Northern Hemisphere!
276-277. ¿What is a “by strong mother trees”? Or do you mean robust mother trees?
339-340. I wonder if “coevolution” is the right word. It sounds for a quite larger time spam than the process described here. Perhaps to say that organic soil formation is a process in parallel and in synergy with the success of regeneration of Larix or so?


Conclusions
Expand the description of the main findings. For example: canopy density, adult tree DBH, aspect, and total N affect the regeneration, but how? More density is benefical or the contrary? Etc.
References

405-407, 408-410. In that reference it seems to me that you provided the given names (Emily, Cornelia, etc.) and not the family names (that are abbreviated) of the authors. It should be the reverse.

---

## Round 0.2 · Major Revisions

I urge you to improve the quality of the manuscript (see 2nd reviewer).

Reviewer 1 ·

Basic reporting

Clear and unambiguous, professional English used throughout.
Yes
Literature references, sufficient field background/context provided
Yes
Professional article structure, figures, tables. Raw data shared.
Yes
Self-contained with relevant results to hypotheses.

Yes

Experimental design

Original primary research within Aims and Scope of the journal.
Yes
Research question well defined, relevant & meaningful. It is stated how research fills an identified knowledge gap.

Yes
Rigorous investigation performed to a high technical & ethical standard.
Yes
Methods described with sufficient detail & information to replicate.

NO. The sampling design is unclear.

Validity of the findings

Impact and novelty not assessed.
Yes
All underlying data have been provided; they are robust, statistically sound, & controlled.
Moderately
Conclusions are well stated, linked to original research question & limited to supporting results.
Yes, although it could improve

Additional comments

The authors made an effort to improve the content taking into account the comments of the other reviewer and myself. Just one more thing. In L.97. On sampling methods. The authors only respond "typical sampling". There is NO sampling method called typical. Namely, there are: Systematic sampling, Stratified sampling, Cluster sampling etc.
It would be valuable for the audience to know the sampling design employed.

Reviewer 2 ·

Basic reporting

The manuscript improved regarding the previous version, but still there are sentences unclear, perhaps because the English language need to be reviewed by a fluent English speaker. Also, there are some too general ambiguous statements, like (Lines from the tracked changes version):

105. “The 14 sampling plots (20 m×20 m) were established by typical sampling.” What do you mean with “typical sampling” ?? ?? It is a too general, ambiguous statement !

282-283. What do you mean with: “regeneration based on statistical theory.” ?? It is a too general, ambiguous statement !

Experimental design

There are some aspects to be clarified in the methodology, like Line 105. “The 14 sampling plots (20 m×20 m) were established by typical sampling.” What do you mean with “typical sampling” ?? ?? It is a too general, ambiguous statement !

Validity of the findings

I guess the findings are in general supported by their data, but it remain unclear some aspects of what they did and how.

Additional comments

I believe the manuscript needs a revision by an English speaker with some knowledge of the general topic. There are expressions that are not completely wrong but neither correct, like to say : (line 97)"The dominant plant species were arbor trees " Does the authors intended to say instead "The dominant plant species were forest trees " ? I guess it is not the best to use "arbor" , and that is a very basic expression for the topic of the paper.

---

## Round 0.3 · Minor Revisions

Please clarify the methodology a bit more. Please improve the English again. I have discovered some more mistakes. I have also included some comments in the attached copy of the manuscript that should be noted.

Reviewer 1 ·

Basic reporting

--

Experimental design

--

Validity of the findings

--

Additional comments

I have nothing more to comment.
The description of the sampling remains unclear. Those 14 plots, were they randomly established. If so, put it. Instead if they decided where to place each of the 14 plots throughout the study area. Then inform the audience. It could perhaps be targeted sampling.

Reviewer 2 ·

Basic reporting

This third (second revised) version improved significatively the English writing, that was needed on the previous versions. Minor clarifications have been made on Methodology, as requested in my previous review.

Experimental design

Minor clarifications have been made on Methodology, as requested in my previous review.

Validity of the findings

Results are supported by the methodology done.

Additional comments

The tracked changed version shows the extensive English writing review done to improve this version.

---

## Round 0.4 · Minor Revisions

I think that the English is still not enough to publish this article.

---

## Round 0.5 · accepted · Accept

Now, this manuscript is ready for publication.